# A Text Segmentation Approach for Automated Annotation of Online Customer Reviews, Based on Topic Modeling

Valentinus Roby Hananto [1,2,*] , Uwe Serdült [3,4] and Victor Kryssanov [3]

1   Graduate School of Information Science and Engineering, Ritsumeikan University, Kusatsu 5258577, Japan
2   Department of Information Systems, Universitas Dinamika, Surabaya 60298, Indonesia
3   College of Information Science and Engineering, Ritsumeikan University, Kusatsu 5258577, Japan;
    serdult@fc.ritsumei.ac.jp (U.S.); kvvictor@is.ritsumei.ac.jp (V.K.)
4   Center for Democracy Studies Aarau (ZDA), University of Zurich, 8006 Zurich, Switzerland
*   Correspondence: valentinus@dinamika.ac.id

**Abstract:** Online customer review classification and analysis have been recognized as an important problem in many domains, such as business intelligence, marketing, and e-governance. To solve this problem, a variety of machine learning methods was developed in the past decade. Existing methods, however, either rely on human labeling or have high computing cost, or both. This makes them a poor fit to deal with dynamic and ever-growing collections of short but semantically noisy texts of customer reviews. In the present study, the problem of multi-topic online review clustering is addressed by generating high quality bronze-standard labeled sets for training efficient classifier models. A novel unsupervised algorithm is developed to break reviews into sequential semantically homogeneous segments. Segment data is then used to fine-tune a Latent Dirichlet Allocation (LDA) model obtained for the reviews, and to classify them along categories detected through topic modeling. After testing the segmentation algorithm on a benchmark text collection, it was successfully applied in a case study of tourism review classification. In all experiments conducted, the proposed approach produced results similar to or better than baseline methods. The paper critically discusses the main findings and paves ways for future work.

**Keywords:** text segmentation; automated multiple-label annotation; online review analysis

## 1. Introduction and Literature Survey

### 1.1. Background

Since its invention a couple of decades ago, topic modeling–a technique based on co-occurrence data–has routinely been applied to reveal thematic patterns "hidden" in diverse and ever-growing collections of texts [1,2]. Besides text documents, topic models were built to classify image [3], video [4], audio [5], and time-series [6] data, and have become an indispensable tool of business intelligence [7,8]. In the e-commerce industry, this technique has been used increasingly to analyze various documents accumulated online. For instance, topic modeling was performed on customer reviews to uncover social and behavioral patterns but also individual experiences associated with consumer goods [9,10], hospitality [11,12], and tourism [13,14]. The successful application of topic modeling to solving these and many other semantic classification problems made it a popular approach to assist or even fully automate the data annotation process [15].

Creating a set of labeled data is a common preliminary task when developing machine learning classifier systems. At the same time, however, producing so-called "gold standard" (i.e., by human experts) training datasets has long been a bottleneck for obtaining high quality classification models. Expert annotations usually require a significant amount of time to be completed and are thus also expensive. Various cost-saving approaches, such as crowdsourcing, have been proposed. However, this approach turned out to produce discrepant and noisy data, and is time-consuming, too [16]. Furthermore, data annotation

relying on human intervention can quickly become infeasible when dealing with a large amount of constantly flowing information [17] that is, however, a typical situation in the e-commerce industry.

As a remedy, various machine learning techniques were proposed to produce "bronze standard" labeled datasets in an unsupervised or semi-supervised manner, minimizing the human involvement [15]. Probabilistic generative models built through topic modeling were utilized to establish semantic membership and assign labels to images [18], social media streaming data [19,20], or textual documents [21]. While being theoretically straightforward, topic model-driven annotation is prone to errors when annotated data include multiple topic-specific segments [22]. Relying on a topic distribution to decide a single label may prevent capturing segment-defined topics, especially in the case of short yet seldom-focused texts, such as online reviews [23].

To break up multi-topic documents into semantically coherent, single label sections text segmentation (TS) would be deployed (e.g., see [24,25]). TS algorithms produce segments (not necessarily topic-bounded) without labels or other semantic information [26]. The latter makes TS alone a poor fit for document automated annotation. While it might appear only natural to try to combine TS with a generative topic model in a context of unsupervised or semi-supervised data annotation, research on the joint application of these two methods remains scarce.

The study described in this paper, therefore, proposes a joint, TS-enhanced topic modeling approach to automated annotation, allowing one to deal with documents which potentially belong to multiple topics. A novel TS algorithm called TopicDiff-LDA is developed to identify topic-bounded segments within a text, based on the topic probability distribution generated with LDA. While breaking multi-topic texts into semantically coherent units, the algorithm works to accommodate for the corresponding changes in class-membership assignments and, consequently, to fine-tune the LDA model. As a result, the updated model provides for a more accurate classification of the segments than with baseline unsupervised methods. The latter makes the developed approach a sound solution to annotate and automatically classify customer-generated texts–requests, feedback, product reviews, etc.–accumulated by online e-commerce systems.

The main contributions of the presented study are as follows:

- TopicDiff-LDA, an original text segmentation method has been developed. The method was experimentally evaluated on a benchmark dataset. Performance was better ($p < 0.001$) than other popular unsupervised TS algorithms tested on the same data.
- A new TS-enhanced topic modeling approach to automated multiple-label text annotation has been proposed. The approach was evaluated in classifier model development experiments. Overall, it demonstrated a better performance than a state-of-the-art semi-supervised annotation method powered by Multilabel Topic Model (MLTM).

The rest of this paper is organized as follows. The next subsection briefly surveys related studies. Section 2 provides for an overview of the proposed approach to data annotation and introduces TopicDiff-LDA, a method for unsupervised text segmentation. Text segmentation experiments are described in Section 3. Section 4 presents a case study of a classifier model for tourism reviews, in which TopicDiff-LDA is deployed to classify and label the training dataset. Experimental results obtained and limitations of the approach are discussed in Section 5. Finally, Section 6 formulates the conclusions.

### 1.2. Related Work

Multiple-label data annotation has long attracted significant attention from the machine learning research community. Zhang and Zhou [27] gave a comprehensive survey of early work in this direction. The authors stayed short of experimental comparison of different methods. They mainly discuss systems built around learning models, such as Support Vector Machine (SVM), $k$-nearest neighbor, logistic regression, and the like. In a related study, Rubin et al. [28] demonstrated that LDA-based approaches generally outper-

form discriminative methods in the case of datasets with many labels. Subsequent studies resulted in an array of probabilistic generative models (e.g., MLTM [29], SMTM [30]), allowing for data annotation in a semi- or "weakly"-supervised mode. As a rule, the developed LDA-based approaches require label-topics or seed keywords to be manually selected in the beginning of the annotation process. The latter may be non-trivial, if feasible at all, in the case of e-commerce data, as multiple-topic manual labeling of large volumes of short texts was shown to produce highly subjective and contradictory results (e.g., see [31]). Wang et al. [32] proposed an optimization algorithm called Adaptive Labeled LDA (AL-LDA) to deal with label disparity. The developed system outperformed many other state-of-the-art multiple-label classification models and worked satisfactory even under massive label noise. An obvious drawback of this approach is its computational cost that would grow enormously in the case of processing large volumes of constantly flowing data.

It should be noted that over the years, several classifier-based systems to assist or automate multiple-label annotation have been developed as well. Largely acceptable results were obtained by applying reinforcement learning (sentence-level labeling of dialogs [33]), deep learning (in combination with LDA [34]) and embedding learning [35]. However, all of these discriminative approaches suffer from drawbacks not unlike the ones pointed to earlier: relying on taken-for-granted human expertise and relatively high computational cost in the case of multiple-label classification. Furthermore, differently from generative models, classifier-based methods provide little, if any, facilities to accommodate for new, previously unseen topics. The latter makes these methods a poor fit for processing dynamic data volumes of, for instance, online customer reviews.

Text segmentation has been used to identify boundaries of semantic units, such as sentences, paragraphs, and news, to mine opinions and emotions, evaluate sentiment, determine language, etc. See [26] for a representative survey of TS applications. Early text segmentation algorithms had a low complexity and often were data- or task-specific, computing segment boundaries based on text similarity metrics (e.g., see [36]). Tagarelli and Karypis [24] developed a method to break a document into coherent segments corresponding to document themes, thus allowing for multi-topic document clustering. Identifying segment topics was left unaddressed. Lu et al. [37] proposed a recursive soft clustering algorithm with built-in topic segmentation for legal document processing. Performance was evaluated in comparison with legal experts. The algorithm utilizes document metadata and requires human involvement to define clusters. In a recent study, Li et al. [38] applied LDA for text segmentation, determining segment boundaries by topic coherence score. As an alternative, Koshorek et al. [39] suggested to learn segment bounds directly from labeled data. The authors used a model pre-trained on Wikipedia texts to break data into semantically consistent pieces. Manchanda and Karypis [25] proposed an iterative algorithm built around a trained classifier to perform text segmentation on multiple-label documents. In each iteration, the algorithm works to refine the segmentation of training data to obtain a more accurate classification model. The authors demonstrated that the approach, while being computationally efficient, achieves similar or better results than popular LDA-based multiple-label classification systems.

Based on the literature reviewed, one would conclude that the high computational cost is the most notable drawback of the existing generative approaches to automated text annotation. On the other hand, the inability to deal with new topics is the most serious limitation of the discriminative approaches. The next section presents an attempt to overcome these problems by deploying classifiers trained on bronze-standard sets. To generate the training data, an LDA model fine-tuned with a novel text segmentation algorithm is used. The proposed approach is partly inspired by the ideas formulated in [32,38], and by the success of the iterative algorithm [25].

## 2. Segmentation-Enhanced Topic Modeling for Automated Text Annotation

### 2.1. Overview

Figure 1 gives an overview of the approach developed in this study. Text documents (e.g., tourism reviews) received for the input are first preprocessed to remove "meaningless" elements, such as stop words, numerals, proper names and pronouns, etc. The text is sliced into words (tokenized), followed by lemmatization of each token to obtain its dictionary form. Part of Speech (POS) selection is performed, with all POS but nouns, verbs, and adjectives being removed. Search for and tokenization of multiword concepts is also performed. All lexical units thus prepared are considered candidate features of topic aspects. The preprocessed tokens are stored in a bag-of-words (BOW) model for training purposes. LDA works with the BOW model to cluster the input documents in an unsupervised manner, assuming that documents with similar content should be grouped together, regardless of their structure. The clusters discovered are interpreted by human experts, using the top 10 to 15 high-probability words in each topic, and class labels are termed. The labels will later be assigned to the documents, based on their class-membership probabilities. Next, the documents get broken into segments by the TopicDiff-LDA algorithm (described in detail in the next subsection). The resulting texts are used to update the topic model, keeping the original labels and the number of clusters intact but re-computing the class-membership probabilities. A set of categorized documents is then produced with labels being assigned, based on segment topicality (i.e., one or more labels per document). Finally, the bronze standard labeled texts obtained with TopicDiff-LDA are used to train a classifier. The classifier model is evaluated and, if found acceptable, is deployed to analyze (classify, sort, label, etc.) other (previously unseen) documents.

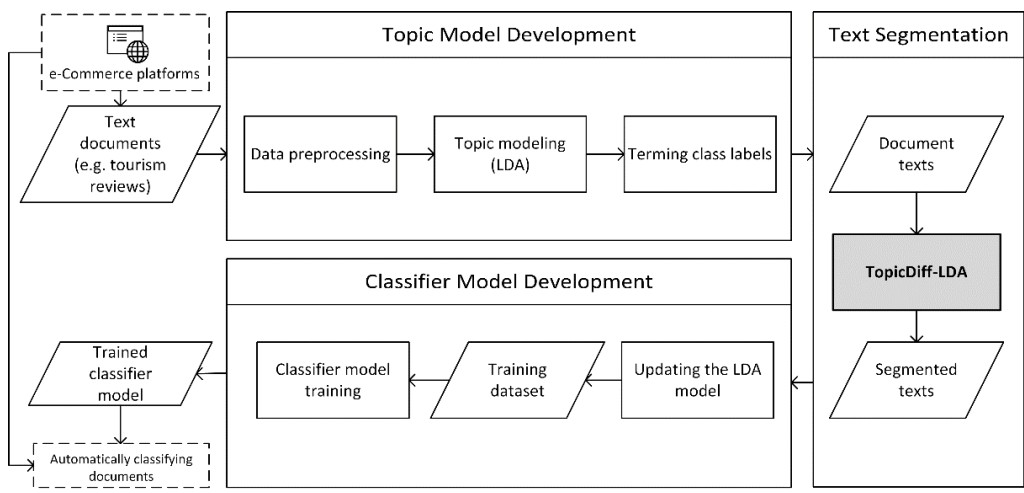

**Figure 1.** An overview of the proposed approach.

### 2.2. Text Segmentation with TopicDiff-LDA

For its input, TopicDiff-LDA is fed $N_D$ preprocessed documents, one document of $n$ sentences a time. The algorithm utilizes the LDA model trained on the whole set of documents, as described in the previous section. The number of segments $N_S$ in the collection is initially equal to $N_D$. To identify topic-specific segments in the document, a sliding window with a minimum segment length of $h$ sentences is implemented. Figure 2 illustrates the application of the window with $h$ set at 3, where rectangles stand for sentences $s_i$, $i = 1, \ldots, n$, and shades indicate the scope of LDA model-based computing in each iteration. The vertical dotted line shows a candidate boundary $b$ (set equal to $h$ in the beginning) of two adjacent segments $segA = [s_1, s_b]$ (the darker shade) and $segB = [s_{b+1}, s_{b+h}]$ (the lighter shade). The size of $segA$ grows (potentially reaching $n - h$) as $b$ is incremented. The size of $segB$ remains fixed (equal to $h$), with its left and right boundaries sliding to the right. Topic probability distributions of $segA$ and $segB$ are generated with the

LDA model, and stored in vectors $X$ and $Y$, respectively. To assess the semantic similarity of $segA$ and $segB$, the Manhattan distance $dist = \sum_{i=1}^{m} |X_i - Y_i|$ is calculated (see [40] for a justification of the selected metric). The vector size $m$ is equal to the number of topics in the trained LDA model (determined by the optimal coherence score $c_v^{opt}$ obtained with an elbow method during the model development; see [41] but also [42]). When $dist$ exceeds threshold $t_{opt} = \underset{t}{\arg\min} \, perplexity_{N_S}(t)$, $b$ is set as the boundary of a new topic-specific segment (i.e., $segA$) and $N_S$ is incremented. The perplexity score for $N_S$ segments in the whole document collection corresponding to a given distance threshold $t$ is computed as $perplexity_{N_S} = exp\left\{ -\frac{\sum_{i=1}^{N_S} \log p(w_i)}{\sum_{i=1}^{N_S} N_{w_i}} \right\}$, where $N_{w_i}$ denotes the number of words in segment $i$, and $\log p(w_i)$ is the log-likelihood of words in the segment. The document processing is terminated when there are not enough sentences left to construct $segB$. The algorithm is run recursively, as specified in Algorithm 1.

**Figure 2.** The concept of a sliding window with $h = 3$ to define segment candidates in TopicDiff-LDA. In each iteration, $b$ sets the candidate segment boundary, so that segments $segA$ and $segB$ (to the left and right of $b$, respectively) can be constructed.

---

**Algorithm 1**: TopicDiff-LDA

---

**Input:** documents, minimum segment size ($h$), LDA topic probability distribution
**Output:** document segments ($DS$)
**Objective function:** $Perplexity(DS_t)$, subject to $\{t \in \mathbb{R}^+, \, t \neq 0\}$
1:     Initialize $t_{seed}$
2:     $DS_{t_{opt}} = $ Search($t_{seed}$) //call Search function
3:     store the best solution: $DS_{t_{opt}}$
4:     **function** Search($t_{seed}$) //returns $DS_{t_{opt}}$
5:     **initialize** $\delta = rand()$
6:     $perpl_{min} = Perplexity($Segmentation($t_{seed}$)) //call Segmentation function
7:     **for** ($i = 1$ *to max_iterations*) **do**
8:       $t_i = t_{i-1} + \delta$
9:       $perpl_i = Perplexity($Segmentation($t_i$)) //call Segmentation function
10:    **if** ($perpl_i < perpl_{min}$) **then**
11:      $t_{opt} = t_i$
12:      $perpl_{min} = perpl_i$
13:      update $\delta$ //determined by the specific optimization algorithm deployed
14:    **return** ($DS_{t_{opt}}$)
15:    **function** Segmentation($t$) //returns segmented documents
16:    **initialize** array $DS_t$
17:    **for each** *document* **do**
18:    $s = $ Sentence_tokenize($document$) //split document into sentences
19:    $DS_t = $ TS($s$, $t$, $DS_t$) //call TS function
20:    **return** ($DS_t$)
21:    **function** TS($s$, $t$, $DS_t$) //returns segmented text
22:    $n = $ Length($s$) //number of sentences
23:    **for** ($i = 1$ *to n*) **do**
24:    $b = i + h - 1$
25:      **if** ($b \leq n - h$) **then**
26:      $segA = [s_1, s_b]$ //define segA

---

```
27:        segB = [s_{b+1}, s_{b+h}] //define segB
28:    X = LDA_infer(segA) //compute the topic probab. distr. for segA
29:    Y = LDA_infer(segB) //compute the topic probab. distr. for segB
30:    dist = Manhattan(X, Y) //Manhattan distance
31:        if (dist > t) then
32:                append segA to DS_t
33:            DS_t = TS([s_{b+1}, s_n], t, DS_t)
34: break
35: else
36: append [s_1, s_n] to DS_t
37: break
38: return (DS_t)
```

## 3. Text Segmentation Experiments

### 3.1. Data

To evaluate the performance of the proposed topic segmentation algorithm, experiments have been conducted, using the Choi dataset [43]. The set was compiled by the author for benchmarking purposes. It contains 700 synthetic documents created by concatenating sentences from different text samples of the Brown corpus. The Brown corpus comprises 374 "informative prose" and 126 "imaginative prose" English documents published in 1961. The Choi dataset is structured into four subsets, depending upon the number of sentences in a segment. Segments were created by randomly choosing a document from the Brown corpus, followed by selecting first $N$ sentences from that document ($N$ selected at random from a predefined range). Each synthetic document consists of exactly ten segments. Segment length varies, as specified in Table 1. Segments are unlabeled, and only segment boundaries are provided in the set description.

**Table 1.** Summary of the data used in the segmentation experiments (document and segment lengths are given in sentences).

| | |
|---|---|
| Number of documents | 700 |
| Number of unique tokens | 5210 |
| Average document length | 83 (1910 tokens) |
| Number of segments per document | 10 |
| Segment length (number of documents) | 3–11 (400), 3–5 (100), 6–8 (100), 9–11 (100) |

### 3.2. Experiments

Eleven LDA models, each with a different number of topics $m$, were trained over 1000 iterations on the Choi dataset. The Tomotopy toolkit was used (https://github.com/bab2min/tomotopy, last accessed on 26 April 2021). The Dirichlet priors were set to $\alpha = 50/m$ and $\beta = 0.01$, as suggested in [44]. With $m$ being incremented from 20 to 220, the optimal coherence score $c_v^{opt}$ was located at $m = 140$ (see Figure 3). The sliding window width $h$ was set at 3, corresponding to the shortest known (or estimated) segment length. Through preliminary experiments it was found that selecting an initial seed value $t_{seed} \sim 2 \times (1 - \mu)$, where $\mu$ is the average highest topic probability of the generated LDA model, results in an improved performance when $t_{opt}$ is searched. For segmentation of the Choi data, $t_{seed} = 1.85$ was used.

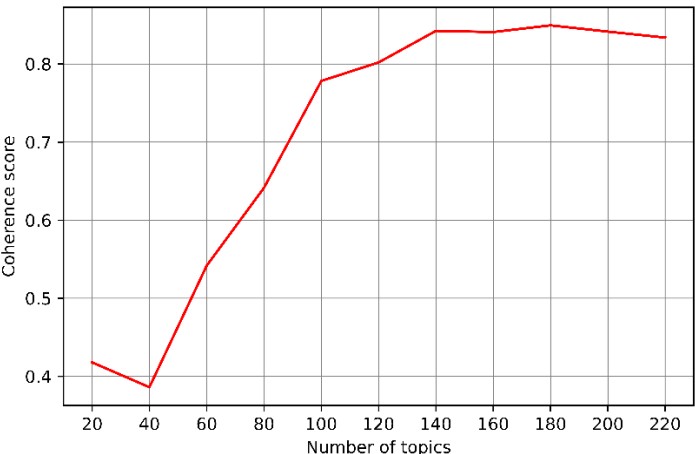

**Figure 3.** Change in coherence score over the number of topics for the Choi dataset.

The TopicDiff-LDA algorithm was run on the 700 documents. The obtained segment quality was evaluated using the *Pk* metric [45]. *Pk* is a probabilistic measure computed by running a sliding window over sentences in the hypothesis (*hyp*) and reference (*ref*) segments. It counts the number of disagreements in sentences being in the same or different sections. Formally:

$$Pk(ref, hyp) = \frac{1}{n-k} \sum_{i=1}^{n-k} \left( \delta(ref_i, ref_{i+k}) \bigoplus \delta(hyp, hyp_{i+k}) \right)$$

where *n* is the number of sentences, and *k* sets a fixed window size calculated as half of the average gold-standard segment size. In a single window, the disagreement of *ref* segment relative to *hyp* segment is calculated as an exclusive disjunction (denoted with ⊕) of the sentence separation $\delta(s_i, s_j)$:

$$\delta(s_i, s_j) = \begin{cases} 1, & \text{if sentences } s_i \text{ and } s_j \text{ are assigned to the same segment;} \\ 0, & \text{otherwise.} \end{cases}$$

Table 2 provides *Pk* obtained for TopicDiff-LDA in comparison with results reported in the literature for the same data (lower scores correspond to better performance).

**Table 2.** Segmentation results on the Choi dataset (unless noted otherwise, text segmentation in an unsupervised mode is assumed).

| Algorithm | Reference | *Pk* |
|-----------|-----------|------|
| C99 | [43] | 0.105 |
| U00 | [46] | 0.078 |
| M09 | [47] | 0.027 * |
| TSM | [48] | 0.009 ** |
| GraphSeg | [49] | 0.066 |
| SegBot | [50] | 0.003 *** |
| TopicDiff-LDA | | 0.029 |

* The algorithm requires pre-training on an external dataset. ** Requires gold-standard set -derived parameter setting. *** A supervised algorithm.

## 4. Case Study

Experiments described in the previous section have demonstrated that the proposed algorithm outperforms other unsupervised approaches but lags behind the supervised and distant-supervised TS methods when run on the artificially created data. However, in a real-life setting of document labeling and classification, there would be little to no opportunity to pre- or re-train algorithms manually every time the processed data is updated. On the

other hand, the latter is a typical situation when dealing with online customer reviews–dynamic and noisy, semantically open-ended yet continually growing collections of texts accumulated by various e-commerce platforms. In [51], the authors presented an attempt to build a knowledge model for tourism recommender systems to automatically classify online reviews. It was found that, while providing for acceptable-quality results on average, LDA-driven single-label annotation often results in multi-topic reviews being misclassified. To address this problem, the TopicDiff-LDA algorithm can be applied, as it was proposed in Section 2.

*4.1. Training Data and The Knowledge Model Used*

Online reviews written in English were crawled from an Indonesian tourism website (https://www.indonesia-tourism.com/forum, last accessed on 31 January 2022). The reviews were manually filtered to exclude uninformative texts (i.e., with fewer than three sentences, written in languages other than English, etc.) A text document collection was thus created, as specified in Table 3.

**Table 3.** The tourism online review collection used to build classifier models.

| | |
|---|---|
| Number of documents | 2685 (filtered from 2807 crawled) |
| Number of unique tokens | 31,781 |
| Average document length, sentences | 12 (278 tokens) |

After preprocessing, an LDA model was built on the review data with a total of 9 topics. The number of topics was decided by observing the coherence score computed for different LDA models but also taking into account general ontology considerations (for more detail, see [51]). By interpreting the top 10 most probable words in each topic, class labels were termed as follows (the topic top-10 tokens are listed after each label):

TOPIC 1. Historical Sites: museum, building, build, house, palace, dutch, mosque, time, collection, old.

TOPIC 2. Protected Area: forest, park, animal, national_park, species, bird, include, plant, conservation, type.

TOPIC 3. Natural Place: cave, river, location, district, road, hill, reach, tree, tourism, meter.

TOPIC 4. Temple: statue, build, stone, side, meter, wall, king, find, roof, base.

TOPIC 5. Mountain: mountain, mount, crater, sea_level, hill, high, peak, regency, view, scenery.

TOPIC 6. Beach: sea, fish, wave, boat, coast, small, beauty, white_sand, reach, sand.

TOPIC 7. General Information: park, travel, tour, want, facility, get, provide, activity, good, offer.

TOPIC 8. Things to Buy: market, batik, food, tourism, product, traditional, plantation, fruit, sell, find.

TOPIC 9. Cultural Heritage: traditional, dance, name, come, hold, become, call, ceremony, day, culture.

Below, this is an example of a review from the processed collection:

*"Watu Dodol Tourism Object in Banyuwangi is located in Kalipuro district, Banyuwangi regency. The location is on Bypass Banyuwangi to Situbondo. The distance from Banyuwangi to Watudodol is 14 km, and from Ketapang port is only 5 km. Watudodol beach usually is full of local tourists for weekends or holidays. The visitors can enjoy the panoramic ocean or stroll to the hill located across the road. From the top of the hill, a beautiful panorama of the Bali strait can be seen. Culinary activities are another interesting thing to do here. Souvenirs made of shells and also stones are on sale in small shops. Arriving at Watudodol from the north route, the Gandrung statue welcomes visitors. This statue is the icon of Banyuwangi; Gandrung is a traditional dance from this*

*city. Located close to Gandrung Statue, there is a big rock that looks like dodol (food made of fruits); probably because of this, the area is called Watudodol. Watu is a Javanese word for rock or stones. There was a mystical story about this rock. The Japanese occupied this area during World War 2, and the Japanese considered this rock distracting their activities. They tried to remove the rock by ordering men to cut the stones, but it did not work. The Japanese then decided to pull it with a boat, and still, it did not work; instead, the boat was drawn. Balinese and also truck drivers are said to put offerings on the rock until today."*

Table 4 presents the topic probability distribution generated with the LDA model for the example. As one can see from the table, there are five topics (3, 4, 6, 8, and 9) with considerably high probabilities. While the narrative goes from "Natural Place" through "Things to Buy" to "Historical Sites," the most likely single label for this text is "Beach" (6), followed by "Natural Place" (3) and "Things to Buy" (8). Assigning (any) one label would obviously be misleading in this case.

**Table 4.** Topic probability distribution of the unsegmented review example.

| Topic | 1 | 2 | 3 | 4 | 5 | 6 | 7 | 8 | 9 |
|---|---|---|---|---|---|---|---|---|---|
| Prob. | 0.002 | 0.001 | 0.252 | 0.129 | 0.001 | 0.318 | 0.013 | 0.150 | 0.130 |

*4.2. Segmentation*

TopicDiff-LDA was run on the review collection with $t_{seed}$ set to 1.0, producing 3562 text segments (a 33% increase from the original 2685 "one-segment" documents). Table 5 shows segmentation results of the review example. The LDA model was updated, and all segments were automatically labeled (one label per segment), based on the re-computed topic probability distributions. Table 6 lists topic probability distributions generated for the example segments.

**Table 5.** The review example segmented and labeled.

| Segment | Text | Label (Topic No.) |
|---|---|---|
| 1 | *"Watu Dodol Tourism Object in Banyuwangi is located in Kalipuro district, Banyuwangi regency. The location is on Bypass Banyuwangi to Situbondo. The distance from Banyuwangi to Watudodol is 14 km, and from Ketapang port is only 5 km. Watudodol beach usually is full of local tourists for weekends or holidays. The visitors can enjoy the panoramic ocean or stroll to the hill located across the road. From the top of the hill, a beautiful panorama of the Bali strait can be seen."* | Natural Place (3) |
| 2 | *"Culinary activities are another interesting thing to do here. Souvenirs made of shells and also stones are on sale in small shops. Arriving at Watudodol from the north route, the Gandrung statue welcomes visitors. This statue is the icon of Banyuwangi; Gandrung is a traditional dance from this city. Located close to Gandrung Statue, there is a big rock that looks like dodol (food made of fruits); probably because of this, the area is called Watudodol. Watu is a Javanese word for rock or stones."* | Things to Buy (8) |
| 3 | *"There was a mystical story about this rock. The Japanese occupied this area during World War 2, and the Japanese considered this rock distracting their activities. They tried to remove the rock by ordering men to cut the stones, but it did not work. The Japanese then decided to pull it with a boat, and still, it did not work; instead, the boat was drawn. Balinese and also truck drivers are said to put offerings on the rock until today."* | Historical Sites (1) |

**Table 6.** The updated topic probability distribution of the segmented review (bold numbers indicate the highest probability for each segment).

| Segment | Topic | | | | | | | | |
|---|---|---|---|---|---|---|---|---|---|
| | **1** | **2** | **3** | **4** | **5** | **6** | **7** | **8** | **9** |
| 1 | 0.006 | 0.033 | **0.539** | 0.033 | 0.005 | 0.327 | 0.009 | 0.036 | 0.007 |
| 2 | 0.006 | 0.004 | 0.146 | 0.107 | 0.004 | 0.241 | 0.008 | **0.344** | 0.136 |
| 3 | **0.334** | 0.004 | 0.209 | 0.004 | 0.005 | 0.008 | 0.282 | 0.007 | 0.144 |

*4.3. Reliability Assessment*

To estimate the reliability of machine labeling, 269 reviews (10% of the whole set) were sampled randomly from the data. Set $A$ with one class-label per review was created by classifying the sampled reviews with a baseline LDA model. This set was subsequently used solely for reference purposes. TopicDiff-LDA was deployed to produce set $B$ with one or more labels per review, using the same data. TopicDiff-LDA also broke the reviews into semantically coherent segments (341 segments in total). Therefore, the number of distinct labels produced by this algorithm for each review can, for comparison purposes, be considered as the number of "perplexity-justified" labels for the given document.

A state-of-the-art semi-supervised multilabel multi-instance algorithm was run to generate an MLTM model (see [29]; the source code was obtained from https://github.com/hsoleimani/MLTM, last accessed on 31 January 2022) that was then used to create set $C$ of multiple-labeled reviews. The labeling procedure assumed assigning as many MLTM labels to each review as the number of segments found in the given text with TopicDiff-LDA. However, the algorithm failed to produce a single label in 14 of the 269 cases.

Two gold standard sets were created. For set $A_g$, each of the 269 unlabeled reviews was manually given a single label by three human annotators. The final label was decided by majority voting. When all three assigned labels were different, one of them was selected at random. The unlabeled reviews were also used to create a multiple-labeled set $B_g$. Three annotators (other than for set $A_g$) were instructed to assign to each review exactly as many not-necessarily-distinct labels as the number of segments determined for the given review with the TopicDiff-LDA algorithm. To resolve annotator disagreements, the same policy as in the case of set $A_g$ was adopted to choose the final labels. Table 7 details the label-structure of the manually annotated sets.

**Table 7.** Structure of the gold-standard labeled sets.

| Class | Average Review Size, Sentences | No. of Distinct Labels Assigned (Avg per Review) | |
|---|---|---|---|
| | | $A_g$ | $B_g$ |
| 1 | 10 | 24 | 26 (1.35) |
| 2 | 11 | 13 | 15 (1.53) |
| 3 | 13 | 55 | 65 (1.48) |
| 4 | 9 | 18 | 21 (1.29) |
| 5 | 11 | 19 | 21 (1.48) |
| 6 | 14 | 67 | 73 (1.42) |
| 7 | 13 | 43 | 57 (1.46) |
| 8 | 17 | 13 | 15 (1.40) |
| 9 | 13 | 17 | 25 (1.68) |
| Total labels: | | 269 | 318 (1.18) |

To assess the level of annotator (dis)agreement in the labeled sets, Cohen's Kappa statistic augmented for multiple labels (see [52]) was calculated. Its pairwise averaged value stays at 0.658 for set $A_g$, and at 0.609 for set $B_g$. The consistency of the machine-labeled sets in respect to human annotation was evaluated with the same statistic. Its values obtained for sets $A$ and $A_g$ (LDA vs. human), $B$ and $B_g$ (TopicDiff-LDA vs. human), and $C$ and $B_g$

(MLTM vs. human) are 0.625, 0.615, and 0.612, respectively. Figure 4 shows the per-class agreement in the multiple-labeled sets.

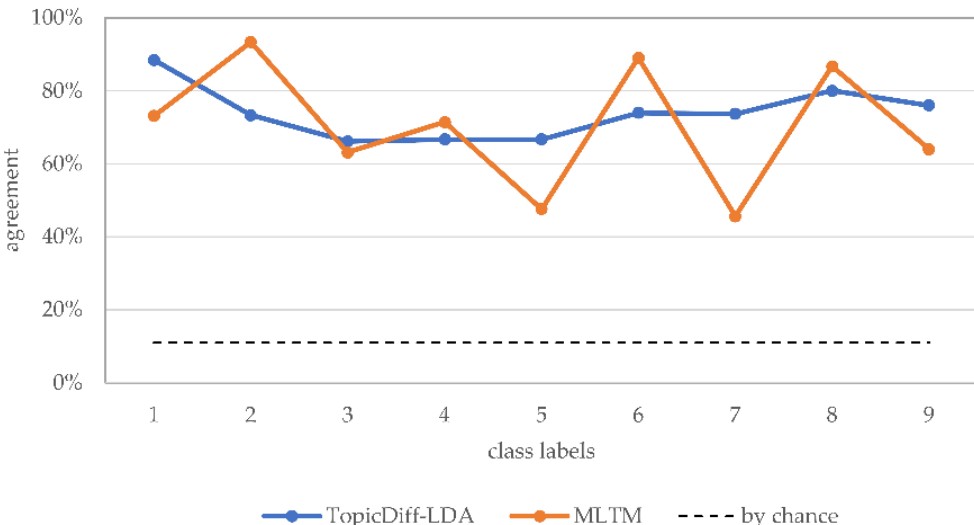

**Figure 4.** Agreement for sets labeled with TopicDiff-LDA and MLTM relative to human annotation.

To compare the classification performance of the multiple-labeling algorithms, the area under the receiver operating characteristic curve (AUROC) statistic was used [53]. The macro (equal class-label weights) and micro (unequal class-label weights) average AUROC values obtained with the data are 0.90 and 0.86 for TopicDiff-LDA, and 0.78 and 0.82 for MLTM, respectively (also see Figure 5).

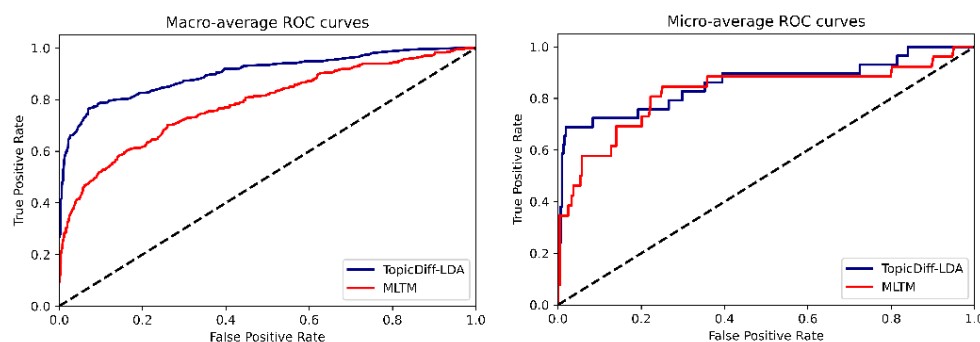

**Figure 5.** Multiple-labeling algorithm performance comparison.

### 4.4. Classifier Model Selection

Two multiple-label bronze standard sets $B_{tr}$ and $C_{tr}$ were created by running TopicDiff-LDA and MLTM, respectively, on the 2685 reviews collected in the study. Four machine learning methods frequently used for short-text classification (see [54])–Convolutional Neural Network (CNN), Random Forest, Logistic Regression, and Linear Support Vector Classification (SVC)–were tested. All the methods but CNN were implemented, using the source code from [55], to deal with multiple-label classification. With $B_{tr}$ set, two different training schemes were used, resulting in two models for each method: one model trained on the whole review texts, and another–trained on the review text segments. (Obviously, with $C_{tr}$, the training could only be performed on the whole review level.) Figure 6 compares the performance of the methods evaluated through 10-fold cross-validation on sets $B_{tr}$ and $C_{tr}$. It should be noted that classifiers trained with a single-labeled set were tested in a previous study but found prone to errors in many cases of automatic review classification [51].

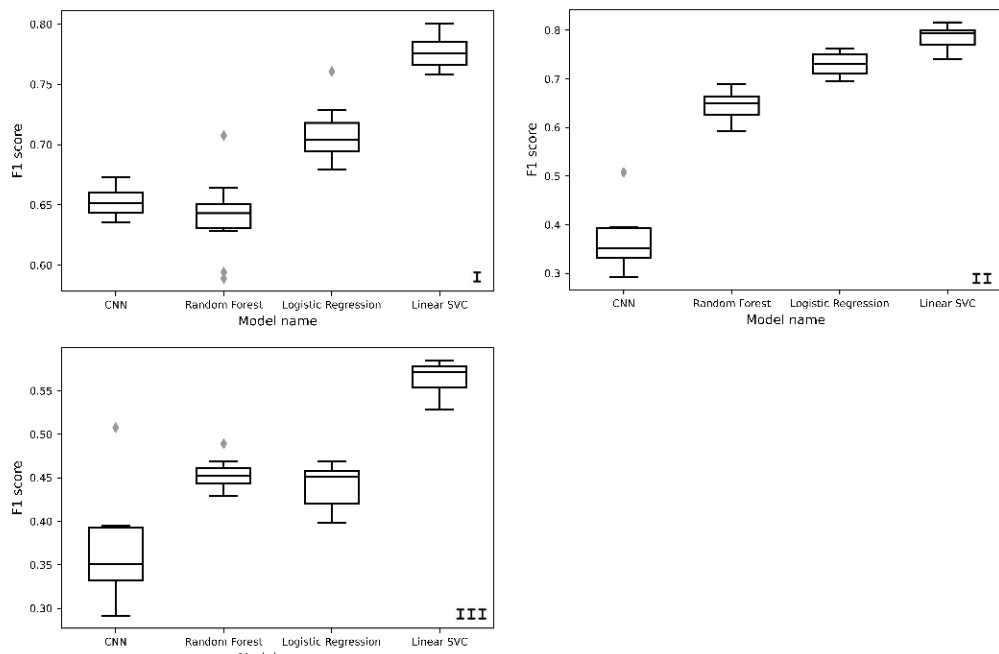

**Figure 6.** F1-score statistics computed through 10-fold cross-validation of the four classifier models trained on (**I**) $B_{tr}$ (review-level), (**II**) $B_{tr}$ (segment-level), and (**III**) $C_{tr}$ (review-level).

Linear SVC performed significantly better ($p < 0.001$) than the other methods in all cases. The averaged macro- and micro-F1 scores achieved the values of 0.78 and 0.78 for the $B_{tr}$ review-level trained model, 0.78 and 0.79 for the $B_{tr}$ segment-level trained model, and 0.56 and 0.60 for the $C_{tr}$ trained model, respectively. Based on the evaluation, it was, therefore, decided to deploy these Linear SVC models for automatic classification and further analysis of tourism online reviews.

### 4.5. Automatic Classification

New, previously unseen tourism reviews were crawled from the same website as in Section 4.1. Three human annotators different from those in the earlier experiments worked to create a gold-standard labeled data set (322 reviews in total, 1.12 labels per review). The same annotation procedure as in the case of $B_g$ (see Section 4.3) was used. The pairwise-averaged value of the augmented Kappa statistic is 0.633 for the new human-annotated set.

The labeled reviews were automatically classified with the three machine learning models pre-trained on the bronze standard data, as described in the previous subsection. To evaluate the classification performance of the models, only those labels were used, on which all three human annotators agreed (226 labels in total). Class-aggregated F1 macro- and micro- scores obtained for the models are as follows: 0.85 and 0.86 for Linear SVC review-level trained on $B_{tr}$, 0.87 and 0.88 for Linear SVC segment-level trained on $B_{tr}$, and 0.76 and 0.78 for Linear SVC review-level trained on $C_{tr}$, respectively. Figure 7 details results of the automatic classification experiment.

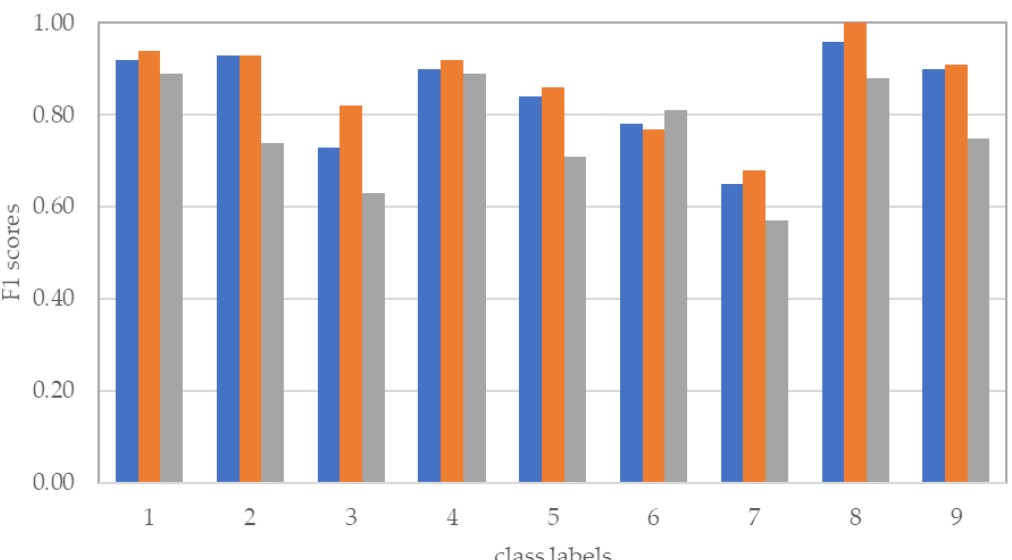

**Figure 7.** Classification performance of models trained on bronze standard sets generated automatically with TopicDiff-LDA (blue and orange bars for training on review- and segment-level, respectively) and MLTM (gray bars).

## 5. Discussion

### 5.1. Segmentation Experiments

Results obtained in the segmentation experiments (Table 2, Section 3.2) convincingly demonstrated the superior ability of TopicDiff-LDA, compared to other unsupervised methods. Also, the performance of the proposed algorithm assessed in terms of $Pk$ improved as the segment length was growing. Segmentation error $Pk$ decreased from 0.36, the value obtained for the set with 3–5 sentences in each segment, to 0.22 for the set with 6–8 sentences per segment, and to 0.16 for when there were 9 to 11 sentences. The same tendency was observed for the segmentation error estimated in terms of WindowDiff ($WD$), a statistic reportedly less sensitive to internal segment size variance [56]: $WD$ changed from 0.22 to 0.21, and to 0.19 as the segment size grew. These results should, however, be expected because longer texts allow for more accurate estimation of the multinomial parameters in LDA [47].

As TopicDiff-LDA relies on the perplexity score to determine segment boundaries (see Algorithm 1), the latter statistic is strongly correlated with the two measures of segmentation error. Assessed with Pearson's coefficient, the correlation ranged from 0.995 to 0.999 for $Pk$ and perplexity score, and from 0.974 to 0.998 for $WD$ and perplexity score on the Choi dataset. Figure 8 illustrates the fact that a segment boundary determined by the minimum of the perplexity score also corresponds to the minimum or near-minimum of the segmentation errors for the data.

Furthermore, Figure 8 reveals that locations of the minimums of the perplexity score and segmentation error measures tend to diverge as the segment size grows. This would be explained by the sensitivity of the perplexity statistic to word likelihood that, with a static lexicon, would be higher for longer texts. TopicDiff-LDA may, therefore, not work well when the expected segment size is much greater than 10 sentences. While the latter is an unlikely scenario in the case of online customer reviews (see [57]), this limitation would hamper the application of TopicDiff-LDA to longer documents.

A well-known drawback of many unsupervised topic-level segmentation algorithms is their high computational cost (e.g., see [47]). To explore the time-efficiency of TopicDiff-LDA, an experiment was conducted by running the algorithm on datasets of different sizes that were randomly sampled from the tourism review set of Table 3. Figure 9 shows results obtained for a computer with an Intel Xeon E5-1650 3.60 GHz CPU and 128 GB DDR4

RAM (all data was segmented, using the same LDA model; the model generation time is not included).

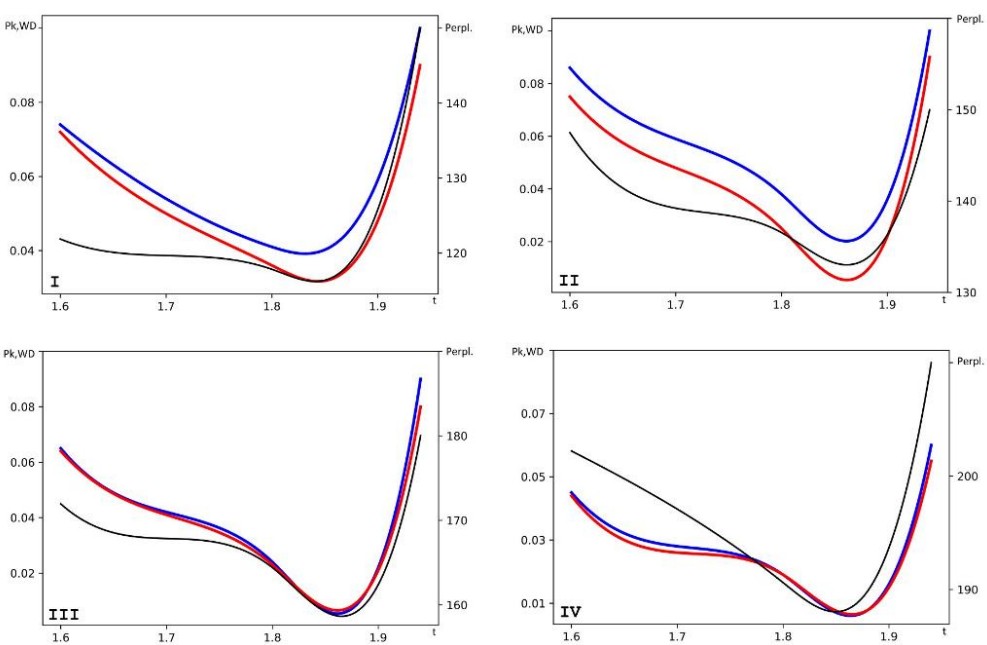

**Figure 8.** As segment boundary is determined by the semantic similarity threshold *t* corresponding to the minimum of text perplexity (black line), segmentation errors *Pk* (blue line) and *WD* (red line) are also minimized. Calculations were performed on the Choi data subsets with segment sizes of (**I**) 3–11, (**II**) 3–5, (**III**) 6–8, and (**IV**) 9–11 sentences. (Note that unlike the perplexity score, *Pk* and *WD*, to be computed, both require gold-standard segmentation of documents.)

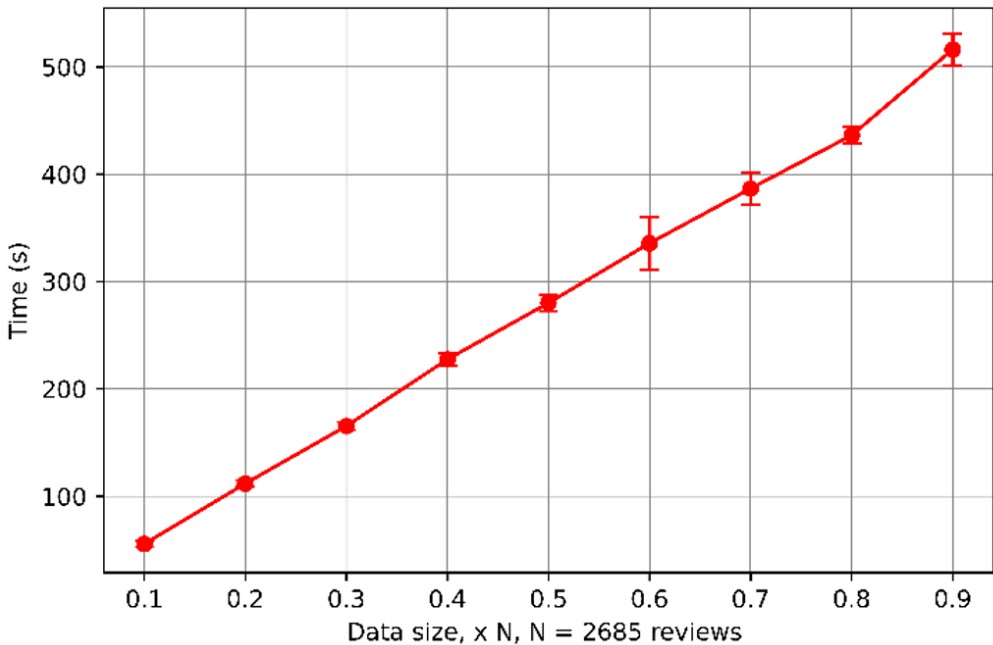

**Figure 9.** TopicDiff-LDA computing time as a function of the segmented review amount. On the graph, each dot indicates the time averaged over 10 random samples of the given size, and the bars show the standard error.

The segmentation time (and, generally, the complexity) of the proposed algorithm is largely determined by the specific implementation of its Search function (see Algorithm 1). In the experiment of Figure 9, this function utilized the standard grid search of a Python library and the segmentation time grew linearly, as the number of reviews increased. For a comparison, some of the best supervised segmentation algorithms also have a linear time-complexity [39]. On the other hand, the runtime of GraphSeg, which is the second-most-accurate among unsupervised algorithms examined (see Table 2), is a polynomial function of the data size [49]. The latter time-complexity holds for many other unsupervised and semi-supervised segmentation algorithms (e.g., [46–48]). Therefore, one would assert that the algorithm proposed in this study has similar or better scalability than the existing baseline methods, whether supervised or unsupervised.

### 5.2. Labeling

The reliability analysis of the gold-standard sets conducted in Section 4.3 confirmed that labeling short online texts is a challenging and often confusing task for human annotators (see [31] but also [58]). At the same time, the inter-annotator agreement achieved in this study for the 9 classes is considerably better than in many other reports which dealt with multiple-class annotation of short texts (e.g., [59,60]). It is understood that the better inter-annotator agreement would be attributed, at least in part, to the fixed (as in [29]) rather than open-ended (as in [59]) labels used in the experiments. The focus of the presented case study was, however, more on automatic classification than on knowledge and label discovery. The results obtained also indicate that for the data, the inter-coder agreement remained nearly the same, regardless of whether single (Cohen's Kappa of 0.658) or multiple (Kappa of 0.609) labels were selected for the reviews. This may be due to the semantic ambiguity associated with interpretation of short and possibly unfocused reviews by annotators lacking contextual domain knowledge. Comparing the final labels with the automated classification results demonstrated that the computer methods tested did not disagree more with the median human assessment (Kappa in the range of 0.612 to 0.625) than the humans disagreed among themselves (Kappa of 0.609) for the data. Furthermore, results depicted in Figure 4 suggest that the human-computer disagreement is not as much dependent on the specific label for TopicDiff-LDA as for MLTM. This may signal that the proposed algorithm is less sensitive to inter-topic variations (in lexicon, scope, etc.) than MLTM. Assessing the overall performances of the two multiple-labeling algorithms (Figure 5), one could conclude that the proposed unsupervised method worked at least as well as, or slightly better than semi-supervised MLTM on the review data. While the considerable difference in the macro-averaged AUROC (0.90 for TopicDiff-LDA vs. 0.78 for MLTM) can be explained by the class imbalance (see Table 7), both algorithms demonstrated "very good" [61] performance in terms of micro-averaged AUROC (0.86 and 0.82, respectively).

### 5.3. Automatic Classification

Results of the automatic classification of the newly collected tourism reviews obtained in the case study suggest that the proposed approach allows for automatically generating bronze standard training sets of an acceptable quality for many practical applications. Performance evaluation of multi-class classification of customer reviews by state-of-the-art application systems trained on gold standard sets typically results in F1-score values ranging from 0.6 to 0.8, depending on the number of classes and the application domain (e.g., see [62,63] and, more generally, [64]). The F1-scores obtained for the TopicDiff-LDA -trained classifiers are in the upper part of this range. The proposed unsupervised approach achieved better classification results than the MLTM semi-supervised method for all but the "Beach" (6) label (Figure 7). By manually inspecting the gold-standard data of this class, it was found that there are several relatively short but multiple-labeled reviews, such as, for example, the following:

*"The Base G Beach (a former American WW2 base) is located about 10 km west of the city of Jayapura, Papua. The beach is beautiful and from here you can look at the Pacific Ocean which is the gateway for ships sailing by from the west. The Base G beach is quiet and still very natural and clean. The water is clear and the beach is made of white sand. The water is so clear you can see clearly through the underwater scenery. Besides enjoying the scenery you can also go swimming, fishing, diving or rent a boat and sail around a bit. Local residents have built some benches and cabins to chill and hang out if you get tired of the sun. There are also several types of trees providing shade. While in Jayapura be sure to visit Base G Beach as it never hurts to spend a day and enjoy this beautiful beach."*

The labels manually assigned to this text by all three annotators were "Beach" (6) and "General Information" (7). The proposed approach failed to assign the "Beach" label as TopicDiff-LDA could not break the review into segments and treated it as a single-label ("General Information") document. One possible reason for that is the minimum segment size parameter set to 3 sentences (the sliding window size, see Section 3.2) in the presented study that prevented the algorithm from detecting shorter segments. Another reason is that TopicDiff-LDA works only with consecutive (i.e., linear and uninterrupted) segments. In the case of MLTM, the minimum labeled unit is sentence, and it could have correctly assigned both labels to the review in focus.

There was practically no impact of the training scheme on the classifier performance. The model trained on (single-labeled) segments did work a little better than the model trained on multiple-labeled reviews, most notably for "Natural Place" (3), "General Information" (7), and "Things to Buy" (8) labels (see Figure 7). This may be due to the classifier being "noised" by the label co-occurrence context in the case of whole-review training. However, the class-aggregated difference is not statistically significant. The latter suggests statistical independence of the classes and, in a sense, validates the choice of labels for the review data.

It is understood that dealing with dynamic, constantly flowing volumes of customer review data would require a transition from static to dynamic topic modeling in the proposed framework. The data of the presented case study was collected for the period of 2011 to 2020–a short time for the tourism domain where knowledge and epistemological perspectives change relatively slowly or, in many cases, not at all. This would not necessarily be true in other domains, such as consumer electronics or politics and governance, where online documents (customer feedback, political party programs, online petitions, etc.) would include new and over-time-correlated topics. The application of the envisaged approach in these domains would require the replacement of the basic LDA in the segmentation algorithm with a method capable of handling dynamic content, e.g., as proposed in [65,66].

## 6. Conclusions

In this paper, an original approach to automatic classification of online customer reviews was proposed. The approach is built around a new text segmentation algorithm developed in the study that is utilized to generate bronze standard multiple-labeled sets for training multi-class classifier systems in an unsupervised manner. The algorithm was tested in various experiments and found suitable, in terms of performance and scalability, to solving the review segmentation problem. It was also deployed in a case study aimed at building an efficient classifier system for tourism online reviews. Results of the case study were scrutinized, and limitations of the approach were formulated.

The presented work offers a theoretical contribution that combines research on unsupervised text segmentation and unsupervised multi-class classification for automated multiple labeling of online reviews. In addition, the proposed algorithm for discovering semantically homogeneous segments in short texts, based on a topic generative model, constitutes a methodological and a practical contribution to the field of advanced computational and linguistic analytics.

In a future study, it is planned to test the developed segmentation algorithm with languages other than English. Modifications of the algorithm will also be examined to incorporate dynamic topic modeling and extend its application to other domains of business intelligence.

**Author Contributions:** Design, methodology, experiments and analysis, writing and editing: V.R.H.; experiments and analysis, writing and editing: U.S. and V.K.; supervision: V.K. All authors have read and agreed to the published version of the manuscript.

**Funding:** No external funding supports this research.

**Institutional Review Board Statement:** Not applicable.

**Informed Consent Statement:** Not applicable.

**Data Availability Statement:** Datasets and Python source code can be obtained from https://github.com/valentinus/TopicDiff-LDA, last accessed on 18 February 2022.

**Conflicts of Interest:** The authors declare no conflict of interest.

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
