# Peer review of "A Text Segmentation Approach for Automated Annotation of Online Customer Reviews, Based on Topic Modeling"

_applsci, doi:10.3390/app12073412_

Round 1
Reviewer 1 Report
This paper is about a text segmentation approach for automated annotation of online customer reviews, based on Topic Modeling. I have the following comments:
- I see some of related work presented in the Introduction section. This is normal but I suggest combining the first 2 sections into one section and name it Introduction and Related Work.
- Please provide a section to describe the limitations of related work at the end
- Regarding the dataset (Section 4.1) is it balanced or imbalanced? If imbalanced, how did you solve this issue?
- Explain more the Choi dataset. For example, what type of documents?
- Provide the limitations of the proposed algorithm
- The authors stated that the proposed approach produced results similar to or better than baseline methods. My question is what about the complexity of the proposed system in comparison with other systems?
- It is recommended to send the manuscript to a professional English reviewer. Some sentences are really long.
Reviewer 2 Report
This paper addresses an interesting and important problem. My main concern is with how the results are evaluated -- the "gold standard" labels. When a human labels an observation as class c, it cannot be counted as wrong for an algorithm to do so as well, unless the definition of the gold standard is that all human reviewers agree. Either use only those labels for which all agree (preferred), or allow for more flexibility. Tables 5 and 6 illustrate this (although these use the estimated rather than human labels). I would argue with the assigned labels for the first two segments. Also, it is implied that the bronze standard labels are only used for training, not testing, but this should be emphasized. Overall the paper makes your case for your method, but the gold standard set needs to be adjusted.
